

# The Parraguirre ice-rock avalanche 1987, semi-arid Andes, Chile - A holistic revision

Johannes J. Fürst[1,★], David Farías-Barahona[1,2,★], Thomas Bruckner[1], Lucia Scaff[3], Martin Mergili[4], Santiago Montserrat[5], and Humberto Peña[6]

★These authors contributed equally to this work.
[1]Institut für Geographie, Friedrich-Alexander-Universität Erlangen-Nürnberg, Erlangen, Germany
[2]Department of Geography, University of Concepción, Concepción, Chile.
[3]Department of Geophysics, Faculty of Physical and Mathematical Sciences, University of Concepción, Concepción, Chile
[4]Institut für Geographie and Raumforschung, Universität Graz, Graz, Austria
[5]Advanced Mining Technology Center, AMTC, Universidad de Chile, Santiago, Chile
[6]Diagua: Derecho e Ingeniería del Agua Consulting, Santiago, Chile

**Correspondence:** Johannes J. Fürst (johannes.fuerst@fau.de)

**Abstract.** Chile is particularly exposed to mountain hazards along the Andean Cordillera. Impact and frequency of devastating debris-flows is expected to increase in the future under climatic warming and urban expansion. To inform monitoring, mitigation and adaptation measures, it is crucial to understand the characteristics of past events in this region. The Parraguirre rock avalanche of November 29, 1987 is a prominent example as it developed into a devastating debris flow reaching 50-km down-valley causing severe damage and loss of human lives. Its destructive power is related to the large water volume involved. The origin of this water is largely unknown - so is the initial trigger volume and the total mass transfer down valley. We therefore retrace the past event using new findings from remote sensing, climate and hydrological records as well as process-based modelling. Important corrections are at order. We find a trigger volume of $17.0 \pm 1.4 \cdot 10^6$ m$^3$ and a total fluid flood volume of $16.0 \cdot 10^6$ m$^3$. The solid mass transfer out of the Parraguirre catchment amounts to $38.1 \pm 15.2 \cdot 10^6$ m$^3$. The high water content cannot be explained by entrainment of soil water and snow cover alone but requires substantial contribution from glacier ice. Furthermore, our simulations corroborate the damming hypothesis of Río Colorado and thereby reconcile the observed wave pulses, arrival times and run-out distance. Apart from the geo-tectonic pre-conditioning, we forward the Parraguirre rock avalanche as a meteorological compound event. The reason is that the spring of 1987 was outstanding in terms of the snowpack height, which preconditioned high snow-melt rates during particularly anomalous warm days at the end of November. Such pre-conditioning is readily accountable in monitoring and early warning procedures for mountain hazards.

## 1 INTRODUCTION

Intensified warming, precipitation extremes, permafrost loss and glacier retreat increase the risk for natural hazards in high-mountain regions (e.g., Gruber and Haeberli, 2007; Biskaborn et al., 2019; Moreiras et al., 2021; Thackeray et al., 2022; Rounce et al., 2023). Steep slopes and high-relief topography make mountain regions prone to massive and destructive mass movements (Huggel et al., 2012; Kargel et al., 2016). Climatic changes exacerbate these hazards (Gariano and Guzzetti, 2016;





Stoffel et al., 2024). Hazards comprise direct events, such as slope instabilities and landslides, as well as secondary events, such as catastrophic lake outburst floods (e.g., Iribarren Anacona et al., 2018; Mergili et al., 2020). Landslides are one of the most common natural hazards in the world and have severe impacts on many components of the high-mountain environment. Run-out distances become particularly large when substantial amounts of water are involved. In this regard, debris flows are a
25 particular destructive type of landslide, as they consist of water-saturated, unsorted non-cohesive material (Hungr et al., 2001; Hungr, 2005)).

The high-relief topography of the Andean Cordillera makes Chile especially prone for debris flows - occurring in all of its climatic zones (Sepúlveda et al., 2006a; Bronfman et al., 2021; Moreiras et al., 2021). Near the capital, a very common trigger
is heavy or persistent rainfall (Sepúlveda et al., 2006b). In this region, the 1987 Parraguirre rock avalanche detached from Cerro Rabicano and developed into a destructive debris flow with a run-out distance of about 50 km (Fig. 1). On its way, more than 29 fatalities were reported and the total damage was estimated to exceed 40 million USD (Casassa and Marangunic, 1993; Hauser, 2002). A detailed description of the debris-flow propagation is presented in Sect. 2. Here, we want to briefly sketch speculations on the main mechanism triggering the initial rock-slab failure. Hypotheses range from geological, via volcano-
seismic, meteorological to hydrological factors. The geological setting near Cerro Rabicano is characterised by sedimentary rocks from the Cretaceous consisting of limestone with intercalations of gypsum and andesite, which are prone for chemical and mechanical disintegration (Hauser, 2002). Disintegration is promoted by almost vertical rock tilt dipping 70°-80° to the west (Casassa and Marangunic, 1993) and periglacial environment processes at these altitudes. The poor mechanical properties of the rocks was undoubtedly a key factor for the initial failure. Volcano-seismic factors have been excluded as explanation
because no important seismic activity was reported in the days preceding the Parraguirre rock avalanche (Eisenberg and Pardo, 1988). In terms of meteorology, 1987 was the 5th rainiest year on a 138-yr record with 712.2 mm in Santiago as compared to the long-term average of 352.7mm (Casassa and Marangunic, 1993). The same is seen for snowfall in the Andes west of Santiago, with 1473 mm water equivalent (w.e.) in 1987 above the long-term 566 mm (Hauser, 2002). It is further known that the days preceding the rock avalanche were particularly warm causing prominent snow melt. In terms of hydrological
effect, surface waters must have quickly been incorporated into rocks through cracks and stratification planes of the limestone sequence. Persistent loss of shear strength of the rock subsurface was a consequence and the resultant gradual weakening is considered another key factor in the debris-flow initiation (Hauser, 2002).

The Parraguirre rock avalanche has received abundant attention in the scientific community (e.g., Eisenberg and Pardo, 1988;
na and Klohn, 1988; Ugarte, 1988; Valenzuela and Varela, 1991; Casassa and Marangunic, 1993; Naranjo et al., 2001; Hauser, 2002; Sepúlveda et al., 2023). These studies are based on field visits, aerial imagery, historical photographs as well as meteorological and hydrological observations. Yet several questions remain to this day unanswered. First, gauge stations report an important flood volume, for which the source remains largely unexplained (Casassa and Marangunic, 1993). Second, existing estimates of the initial rock-slab volume show large discrepancies (Naranjo et al., 2001; Hauser, 2002). Third, no estimate of
the total debris-flow volume was reported - let alone its partitioning into its fluid and solid portion. Fourth, the multiple waves,





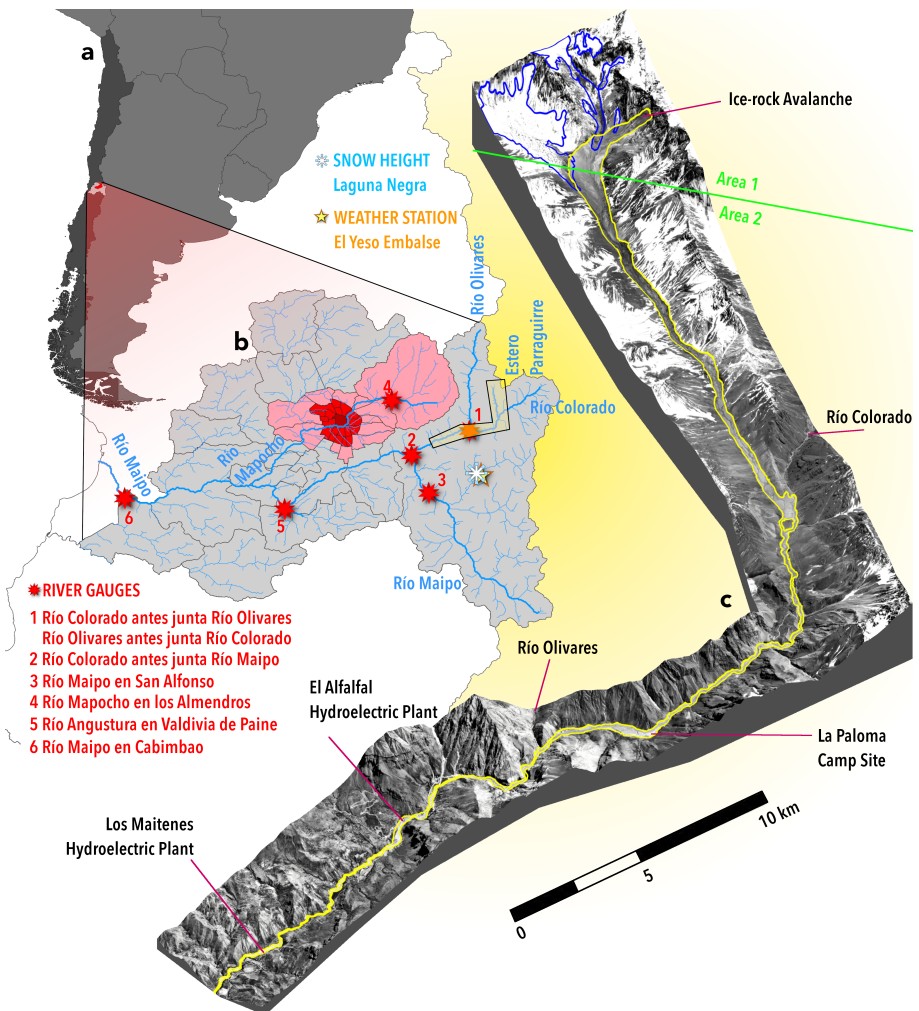

**Figure 1. Overview of Estero Parraguirre and Río Colorado catchments.** Southern South America (a) and inset of the Metropolitan Region of Santiago de Chile (b). Different red shading indicates the commune area of Santiago. Main rivers in the river system (blue) are highlighted. Also shown are river gauges (red stars), a weather station (yellow star) and snow measurements (snowflake symbol) with coverage in 1987. The yellow shaded polygon indicates the extent of the Parraguirre debris flow event. The 3D representation (c) is produced from 1987 aerial images. The yellow outline indicates the observed impact area. Blue outlines represent glacierised areas. For the r.avaflow simulations, friction coefficients are calibrated in two distinct areas (divided by green line, area 1 & area 2 in green font). Aerial photos (c): credit & courtesy of SAF, Chilean Air Force, 1987.

which were observed, nourished the hypothesis that the debris flow did temporarily stop and was re-active after dam breaches. This hypothesis remains both unchallenged and unconfirmed.



Here, we pursue a first attempt to answer these open questions with regard to the 1987 Parraguirre debris flow by combining the existing records with hydro-climatological data, remote-sensing observations and process-based modelling. In the following, we first re-draw the event history by compiling the available information (Sect. 2). Thereafter, we present the utilised data sources from remote sensing as well as from atmospheric and hydrological monitoring (Sect. 3). Subsequently, the applied analysis methods and model experiments are specified (Sect. 4). Dedicated sections then present our results (Sect. 5), which are carefully put in perspective (Sect.6).

## 2  HISTORY OF THE LANDSLIDE ON NOVEMBER 29

### 2.1  Initial Rock Fall and Ice-rock Avalanche

On 29 November 1987, a massive ice-rock avalanche occurred in the Central Andes of Chile (Fig. 1, Table 1). The avalanche originated from the northwestern slopes of Cerro Rabicano (33.3236°S, 70.0027°E; Piderit, 1940; Ambrus, 1967) at an altitude between $4000$ and $4500$ metres above sea-level (m a.s.l.) (Fig. 2c). The avalanche descended westwards and hit the valley bottom ($3500$ m a.s.l.) at 10:33am (Casassa and Marangunic, 1993; Hauser, 2002). The timing is well known as the impact was recorded by several seismographs in the Maipo catchment (Eisenberg and Pardo, 1988). At about the same time, a dust cloud was visible to the southeast from Farellones – at a distance of 25km (Casassa and Marangunic, 1993). Estimates of the initial volume of this rock avalanche range from 5 - 6·$10^6$ m$^3$ (Valenzuela and Varela, 1991; Casassa and Marangunic, 1993; Hauser, 2002). Previous studies have speculated that the average thickness of the initially slab of rock, that detached, falls between 20 and 40 m. Also, the release area was controversially discussed. Casassa and Marangunic (1993) forwarded 0.1 km$^2$, whereas Hauser (2002) suggested 0.5 km$^2$. The impact area is located in the headwaters of Estero Parraguirre at the valley head. From there, the landslide first overran a glacier tongue (Glacier N°24) and then climbed a 50-m ridge before it followed a sharp 90°-turn south into the Estero Parraguirre. To overcome the ridge, minimum speeds of 31 m s$^{-1}$ were necessary (Hauser, 2002). Observed mudline marks on both valley sides of the 90°-turn suggest similar speeds of 24 m s$^{-1}$ (Casassa and Marangunic, 1993). It is further assumed that, at the time of failure, snow and ice melt was elevated resulting in saturation of the deposits that formed the valley fill (Hauser, 2002). As a consequence, the initial rockfall was quickly transformed into a powerful debris flow likely within the first 5 km of its source.

### 2.2  Debris Flow in Estero Parraguirre

The debris flow then pursued its way southwards along Estero Parraguirre. Along its path, it was suggested that it incorporated considerable amounts of solid material and water (Casassa and Marangunic, 1993; Hauser, 2002). From aerial photographs acquired a few days after the event, a patchy snow cover is visible up to 11km away from the source (Fig. 2e). Mud splashes and superelevation of the debris line along the valley confirm the high mobility of the landslide in this part - evidence for high water saturation.



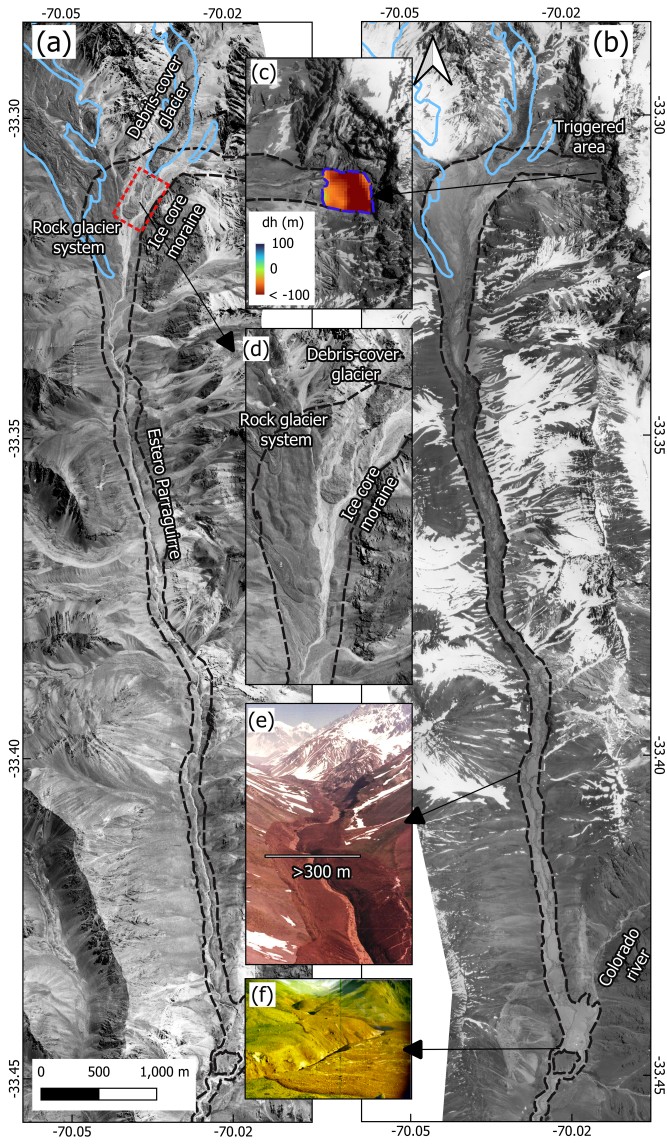

**Figure 2. Orthomosaics and DEM reconstruction of pre- and post-event.** (a) Orthomosaic from 1955 depicting pre-event conditions where dash black line is the impacted area. (b) Post-event orthomosaic showing the impacted area. (c) Elevation changes (in meters) in the triggered region. (d) Zoomed-in view of panel (a). Several glacial landforms are visible, including a debris-covered glacier (Glacier N°24) and an ice-cored moraine near the 1955 glacier terminus. Additionally, a rock glacier system (Glacier N°22) is observable, captured five days after the event. In-situ photographs (looking north) showing the debris flow path and its consequences in the Estero Parraguirre (e) and at the confluence (f). In-situ photos (e, f): credit & courtesy of Humberto Peña, 1987. Aerial photos (a-d): credit & courtesy of IGM Chile, 1955, and SAF, Chilean Air Force, 1987.





One kilometre before reaching the confluence with Río Colorado, the primary debris flow left the narrow riverbed of Estero Parraguirre and overran 10 to 15-m high fluvial terraces towards the east (Fig. 2a). This overspill occurred along 100-400 m and across a width of 600-800m leaving deposits of as much as 3m height (Casassa and Marangunic, 1993), finally discharging into Río Colorado. Speculations on the immense flow volume in this area nourished the hypothesis of a temporary damming of Río Colorado (Hauser, 2002)). A smaller secondary flow followed the Parraguirre channel and joined Río Colorado some distance downstream. The high energy of the debris flow in the confluence area is illustrated by the displacement of a huge 1000 m$^3$ boulder. While presumably located upstream of the confluence prior to the debris flow, the boulder was retrieved several kilometres down-valley (Casassa and Marangunic, 1993; Hauser, 2002).

### 2.3 Debris Flow in Rio Colorado

Down-valley of the confluence, the debris flow got diluted in the Rio Colorado. A truck driver observed the flow event and estimated a 10-m yr$^{-1}$ propagation speed. Along the following section, observed superelevation of debris lines suggest maximum flow-heights of about 4 m, on average. In bends, mud splashes however reached 40 m high, indirectly reporting on the flow ferocity. In this section, maximum deposition heights of 3 m are reported (Casassa and Marangunic, 1993).

The next evidence for the progression is from 11:20 (Table 1) at about 27.5km from the source (Valenzuela and Varela, 1991), where the debris flow reached and destroyed the construction camp near Estero La Paloma. At La Paloma, the narrow Colorado valley (50-100 m) opens again into a larger plain of 200-300m width. There, average deposition heights were reduced to about 0.6m (Hauser et al., 2002).

In the literature, there is some timing confusion between the El Alfalfal powerhouse and the Los Maitenes hydroelectric plant (cf. 1). We follow Hauser (2002) and assume that a first small wave hit Los Maitenes around 12:14. This wave alerted many people (shut-down and evacuation) before the second more destructive wave hit the facilities at 12:37. During this event, maximum flow heights in the river channel reached 30m. The thickness of debris-flow deposits along this section ranged between 2 - 4 m (Casassa and Marangunic, 1993), with a typical value of 2.5m (Hauser, 2002). These deposits typically consisted of unstratified mud-rich clast conglomerates (42-44% fine grained) bearing sandstones, red shales, andesites, magnetite-bearing dacites, limestones and gypsum (Naranjo et al., 2001). Downstream of Los Maitenes and towards Rio Maipo, no significant damage was reported as the debris-flow became increasingly diluted, the riverbed widened and the slopes reduced. In a 50-m wide gorge before Río Maipo is reached, flow heights of 4m were observed.

### 2.4 Debris Flow in Rio Maipo

Unfortunately, the gauge station just downstream of the Colorado-Maipo confluence (El Manzano) was destroyed and no measurements are available from November 3 until May 27. The station at Cabimbao remained operational and recorded the full event on November 30 (Fig. 1). Two discharge peaks were reported with the main one at 07:00 being 175 m$^3$ s$^{-1}$ above a baseline value of about 700 m$^3$ s$^{-1}$ (na and Klohn, 1988). From this a total flood water volume of 7·10$^6$ m$^3$ was estimated.



Casassa and Marangunic (1993) speculated that this water volume had to be mobilised already in the Parraguirre catchment. Accounting for an extensive 1-m snow cover ($3.5 \cdot 10^6$ m$^3$), pore water in the rock slab ($0.1 \cdot 10^6$ m$^3$) and saturated soil ($0.6 \cdot 10^6$ m$^3$) along the valley bottom, about half of the observed flood volume can be explained. This 50%-value is called into question
because no closed snow cover was observed during the helicopter survey a day after the event (Fig. 2).

## 3 DATA

### 3.1 Digital Elevation Models, Orthomosaics & Reconnaissance Photographs

We have produced two digital elevation models (DEMs). The first DEM was derived from 1200dpi scans of the topographic maps (1:50.000) generated by the Instituto Geografico Militar (IGM). The maps were created and georeferenced from aerial
photographs taken in 1955 (HYCON) over Central Andes of Chile. The maps were originally referenced horizontally to the PSAD56 datum with a UTM projection in zone 19S. The vertical reference is based on orthometric heights (in m a.s.l.) derived from precise trigonometric levelling using theodolites from tide-gauge benchmarks located along the coast. To generate the 1955 DEM, we digitalised the contour lines and georeferenced to WGS84 using standard GIS procedures to reproduce a 30m resolution DEM representative of the topography before the event.

The second DEM was generated from aerial photographs taken a few days after the event. The aerial photographs, which were scanned photogrammetrically, were provided by the Chilean Air Force's Aero-photogrammetric Service (SAF). The digitised images were orthorectified and georeferenced using structure from motion (SfM) techniques. For georeferencing, ground control points of non-glacierised sites were extracted from the 1955 DEM (IGM map) and PlanetScope satellite imagery (Planet
Labs Research and Education Initiative). Based on these GCPs, the SfM workflow comprised an iterative routine of block bundle adjustments (BBA) and refines camera calibration parameter adjustments. We obtained a 30-m resolution. We obtained two orthomosaic image from aerial photographs pre and post event (Fig. 2), which allows us to further define the trigger and impacted area of the Parraguirre debris flow.

The day following the event, the Water Directorate (DGA) conducted a reconnaissance flight to assess the extent of the damage along Estero Parraguirre and Río Colorado. A series of in-situ photographs taken during this flight were made available to us, providing valuable data to enhance our interpretation of the triggered ice-rock avalanche and the affected area. These photographs not only helped in identifying the impacted zones more accurately but also allowed us to better understand the magnitude of the damage caused by the event. The visual evidence gathered from the field photographs complemented our
remote sensing data and field observations, giving us a more comprehensive view of the geomorphological changes and the potential downstream impacts on infrastructure.





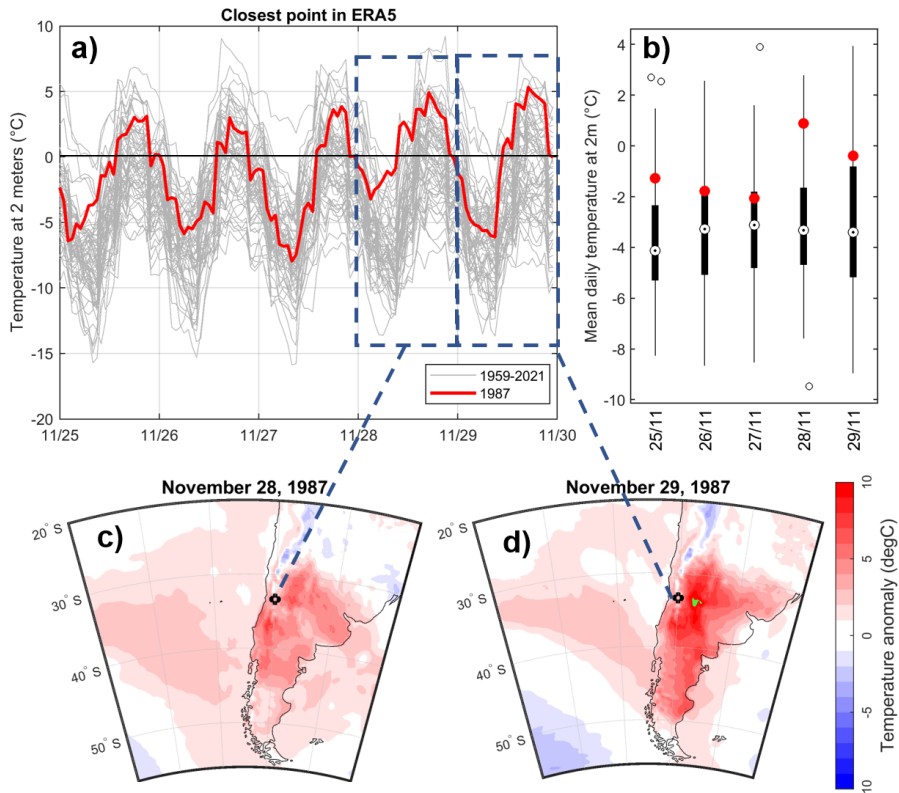

**Figure 3. Reanalysis air temperatures.** Hourly air temperature from reanalysis data between November 25 and the 30 for 1959 - 2021 (a). Box plots of the 63 years of daily air temperature between November 25 and 29. Red dots indicate the year 1987. Panels (c) and (d) show the air temperature anomaly in a synoptic scale on November 28 and 29.

## 3.2 Meteorological Data

The first source for meteorological information is station data from El Yeso Embalse (70° 5' 19"W, 33° 40' 33"S, national station ID: 330149). The station is located at $2\,475$ m a.s.l. about 20 km south of Río Colorado ( 40 km from trigger area). The
Chilean Water Directorate (Dirección General de Aguas de Chile; DGA) provides daily temperatures and precipitation data from 1977 to 1993 and 1962 to 2020, respectively. Other DGA station data on snow height is available in the Maipo Canyon. These snow surveys are one of the most reliable measurements of water availability in the Chilean mountains. For this study, we analysed the observed snow water equivalent (s.w.e.) at Laguna Negra (70° 6' 28"W, 33° 39' 57"S. national station ID: 330146) within 2 km distance (NW) of the weather station at El Yeso Embalse.

Apart from in-situ observation, reanalysis data from ERA5 (Hersbach et al., 2020) is selected to evaluate the synoptic and meteorological conditions during the event. ERA5 provides atmospheric data at about 30 km grid spacing for the entire globe



in hourly time steps. In this study, we analysed ERA5 from 1959 until 2021 during months between September and December, which is considered the season when snow melts start. The data is used to identify anomalies in daily mean and maximum air
temperatures at two metres above ground, and the daily snow depth and the maximum snow depth in the accumulation season anomalies.

## 3.3 Hydrological Data

Gauge-station data is publicly available from the Chilean Ministry for Public Works (Ministerio de Obras Públicas, MOP) and more specifically from the DGA directorate. Time series of river discharge observations were retrieved for altogether 7 stations
in the Maipo catchment area (Fig. 1): Río Maipo at Cabimbao, El Manzano and San Alfonso, Río Colorado (before Olivares & before Maipo), Río Olivares, Río Mapocho (Almendros) and Río Angostura (Valdivia en Paine). These stations were selected because continuous data was available in 1989 (except for Río Colorado & Olivares).

## 4 METHODS

### 4.1 Remote Sensing & Photogrammetry

To estimate the total release volume and elevation changes in the impacted area, the DEMs were initially resampled and co-registered both horizontally and vertically using the method developed by Nuth and Kääb (2011). This co-registration process ensured a high level of cross-alignment and effectively eliminated outliers. The elevation changes from the trigger point to the impacted area were then determined by subtracting the 1987 DEM from the 1955 topographic DEM. These elevation changes serve to quantify the initial trigger volume and the down-valley mass transport. To calculate volume uncertainties, we follow
Farías-Barahona et al. (2019). In this way, we account for uncertainties in the DEM differencing over stable areas, and for uncertainties in areal extents.

To quantify the impacted and triggered areas, we manually digitised both regions from the mosaic generated using 1987 aerial photographs, following standard procedures. The manual digitisation was supported by in situ photographs taken in the days
following the event, which allowed us to photo-interpret specific areas.

### 4.2 Multi-phase Mass Flow Modelling

#### 4.2.1 r.avaflow

We retrace the mass movement in Estero Parraguirre with the multi-phase mass-flow model r.avaflow (version 3.0), an open-source model integrated into GRASS GIS as a raster module (Mergili et al., 2017). In general, r.avaflow computes the prop-
190 agation of rapid mass-movements and is capable of simulating a mixture of fluid, fine-solid, and solid phases (Mergili et al., 2017, 2018a, b; Pudasaini and Mergili, 2019; Mergili et al., 2020; Vilca et al., 2021). We only employ the fluid and solid phases capabilities. The fluid phase represents a mixture consisting mainly of water and very fine particles, e.g., clay and silt. Its de-





formation is described as non-viscous and shear-rate dependent. The solid phase contains the largest grain sizes like boulders, cobbles and gravel. Deformation is assumed non-frictional and shear independent. The model solves for mass and momentum

conservation in an efficient way, relying on the Total Variation Diminishing Non-Oscillatory Central Differencing Scheme (Wang et al., 2004). Momentum transfer between the phases considers viscous drag, buoyancy and virtual mass induced by acceleration differences. This transfer is key to produce concurrent deformation, mixing, and separation of the phases (Mergili et al., 2017).

Integral input variables and decisive control parameters are provided as raster-maps. These comprise a digital elevation model (DEM) of the pre-event situation, the fluid and/or solid release volumes, maximum entrainment heights and solid entrainment fraction as well as internal and basal friction coefficients for the solid phase. For further customisation, r.avaflow allows for timed volume release rates in form of phase-specific hydrographs placed at predefined locations. For validation, diverse scores are computed to assess the accuracy of the modelled mass movement against the observed impact area. Scores are computed

from the percentage of correctly and incorrectly identified points.

### 4.2.2 Experimental setup

In all experiments, a 30-m resolution is used and the release volume is prescribed from the observed elevation change giving $17 \cdot 10^6$ m$^3$. Moreover, the region is divided into two sub-domains: Area 1 and Area 2 (Fig. 1). In each region, specific values are set for the friction coefficients, the maximum entrainment height and the solid entrainment fraction. Entrainment is constrained

to the observed impact area.

*Calibration Experiment 1 (CAL1): Entering Estero Parraguirre*

In this first set of experiments, the target was to infer optimal friction parameters such that the debris flow does follow the sharp 90°-turn southwards into the main valley of Estero Parraguirre (Fig. 1). For our assessment, we used the match with

215 the observed impact area, and that the simulated debris flow does not overrun the glacier in the west. For this purpose, the internal (PHI1) and basal friction (DELTA1) parameters for the solid phase were varied according to Table 2. Furthermore, the maximum entrainment height (HENTR1) along the valley bottom (Area 1) was prescribed from the observed elevation change. The rectangular domain spans the trigger site and La Paloma camp site.

*Calibration Experiment 2 (CAL2): Parraguirre – Colorado – La Paloma*

The primary targets in this second calibration step are threefold. First, we require the resultant debris flow to reach the confluence with Río Colorado and, according to the reported timing, to arrive in time at La Paloma camp. Second, we assume that the fluid volume exiting Estero Parraguirre fully explains the flood volume inferred from Cabimbao gauge station. The specific target value is presented in the Results Sect. 5. The third calibration target, relates to the net solid mass export out of the Estero

Parraguirre catchment as inferred from observed elevation changes (Sect. 4.1). For the calibration purpose, friction parameters were adjusted in Area 2 following Table 2. For the admissible range in these target values, please refer to the results section.





The spatial domain is the same as in CAL1.

*Calibration Experiment 3 (CAL3): Damming Experiments*

The admissible parameters after calibration in CAL2 enter our final simulations, for which a damming of Río Colorado is prescribed. We assume above average river discharge of about 28.0 m$^3$ s$^{-1}$ upstream of the confluence with Estero Parraguirre. We infer this number from a simple scaling argument. Hauser (2002) reported typical November discharge values of Río Colorado above the confluence with Estero Parraguirre of 13.3 m$^3$ s$^{-1}$. As data from the gauge station at Colorado upstream of the Maipo confluence is available until November 28, we compute a monthly mean and compare it to the multi-annual average (cf.

Table 3). The resultant scaling factor is 2.1. We estimate the damming time to be about 30 minutes. In the model, the dammed water volume is released at 11:10 (2220s) for a duration of 1, 2, 5 or 10 minutes at constant rates at a location just upstream of the confluence (33.42934˚S, 70.01346˚W). This breaching time is chosen such that the main wave of the debris-flow reaches La Paloma around 11:20. For these simulations, deposition and stopping of the debris flow were deactivated. The timing choices are motivated from the results of CAL2 and explained in the results (Sect. 5). The domain size spans a rectangle from Los

Maitenes to the trigger site.

## 5   RESULTS

### 5.1   Mass movement and volume estimation

Our results reveal a triggered volume of $17\pm1.4\cdot10^6$ m$^3$ impacting a total area of 12 km$^2$. Figure 2 illustrates the affected region

in the upper Parraguirre before and after the event, highlighting the extensive mass movement that occurred. This triggered volume significantly impacted both the debris-covered glacier and a rock glacier in the vicinity. But also impacted an ice-core moraines formally connected to the debris covered glacier. The contribution from the ice melted from these landforms played a crucial role in the liquid phase of the subsequent debris flow, enhancing its mobility and destructive potential. Furthermore, we estimate that the mass transfer out of the Parraguirre catchment was $38.1\pm15.2\cdot10^6$ m$^3$, leading to significant landscape

alterations. In several sections, the displaced material has notably altered water flow, which in turn has considerably increased sedimentation in downstream areas.

### 5.2   Pre-event Meteorological Conditions

The last week of November of 1987 the synoptic conditions showed an intense Eastern Pacific Anticyclone (high pressure system at surface) off coast of Chile along with strong southerly flow parallel to the mountains, following the anticyclonic

circulation (Suppl. Fig. S2a). It appears that the reanalysis was able to capture a large scale-forced easterly flow across the upper part of the Andes near Central Chile (Suppl. Fig. S2b). The associated downward flow across the eastern mountain slopes, produced Foehn-type conditions. It implies warming on the descending wind (subsidence) around 700 hPa and thus





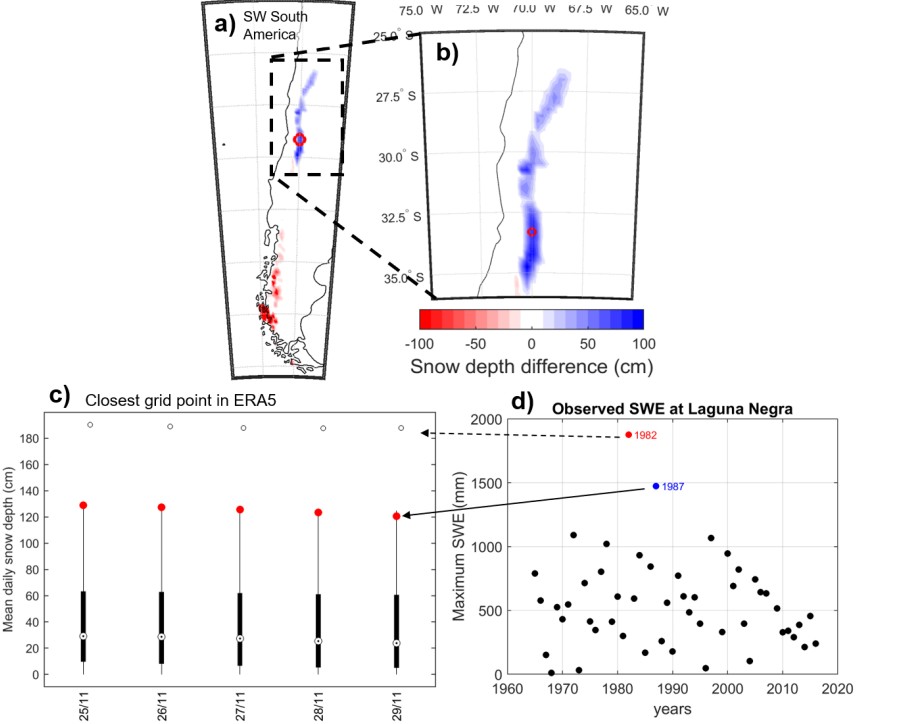

**Figure 4. Snow depth information from reanalysis and station data.** Snow depth anomaly over south-west South America from the reanalysis ERA5 (a, b). Panel (c) presents a boxplot of mean daily snow depth for the 63 years (1959 to 2021) and the year 1987 is highlighted in red. Panel (d) shows the Laguna Negra Snow Water Equivalent (SWE) observations. The years 1982 and 1987 are highlighted (red & blue).

intensifying the warming at high elevations at the Andes.

The meteorological conditions resulted in a positive temperature anomaly of the reanalysis over most of Southern South America (Fig. 3c, d). In addition, it is clear that the day before the avalanche (November 28), the minimum temperature was especially high (Fig. 3a). Absolute values are more difficult to interpret. We therefore consult the temperature observations at high elevation, i.e., at the El Yeso Embalse weather station (Suppl. Fig. S3). There, daily near-surface air temperatures exceeded zero degrees for the entire October and November. This station is at a similar elevation as the Estero Parraguirre (Table 1). After a

step-increase in daily temperature of 5-10°C between November 5 and 10, snow melting did start at the trigger site ( $\sim 4\,200$ m a.s.l.). After some warmer days between November 14 and 22, an extraordinary warm spell was experienced on 28. and 29.11.1987 with average daily temperatures exceeding 5°C. The warm anomaly on November 28 is also confirmed in ERA5 (Fig. 3a). Daily mean temperatures of 5°C can cause snow melt of up to 3-10 cm per day (considering ranges for snow density of 300-500 kg m$^{-3}$ and for snow-melt degree-day factors of 3-7 10$^{-3}$ m w.e. °C$^{-1}$ day$^{-1}$).





Considering precipitation measurements, no rainfall was recorded after October 15, more than a month before the debris flow (Suppl. Fig. S3). However, snow-height observations and reanalysis data indicate a particularly high snowpack in 1987 over the Andes in Central Chile (Fig. 4a,b) from a wet winter season. This likely produced the unique antecedent conditions for the avalanche on November 29. Looking at the closest point of ERA5 (at 3 906 m a.s.l.) to the avalanche location (Fig. 4c) and the

observations of maximum Snow Water Equivalent (SWE) at Laguna Negra (30 km south of the Parraguirre creek headwater), the snow depth showed clear positive anomalies in 1982 and 1987 (Fig. 4d). Looking in more detail in the year 1982, a clear high snowpack is recorded and simulated by ERA5. However, in 1982 there was a cold spring, without any positive temperature anomaly to trigger rapid snow melt (Fig. 5).

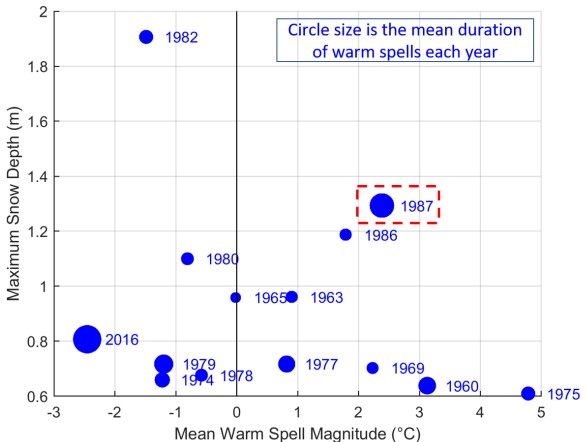

**Figure 5. Maximum snow depth, mean warm spell magnitude and duration in spring.** This figure exclusively displays the years that contain a warm spell during spring. The circle diameter scales with the mean duration of warm spells.

To highlight the extraordinary meteorological conditions of 1987, we quantify the warm-spell magnitudes and durations for each year over the ERA5 historical data (from 1959 to 2021) as well as the maximum snow depth (Fig. 4). All three quantities show elevated values in 1987 (Fig. 5). For the entire record, we find the second largest snowpack, the second longest duration of warm spells and the third warmest warm-spell magnitude.

## 5.3 Hydrological Conditions Preceding the Parraguirre Ice-rock Avalanche

The typical November situation of the river discharge in the Maipo system is characterised by about 150 m$^3$ s$^{-1}$ near the ocean outflow (Table 3). The main input (∼70%) originates from the headwater of Maipo itself. Sorted by importance, additional input comes from Colorado, Angostura and Mapocho. November values in 1987 show a more than twofold increase in all gauge stations. This appears highly atypical because these extremes exceed two standard deviations.



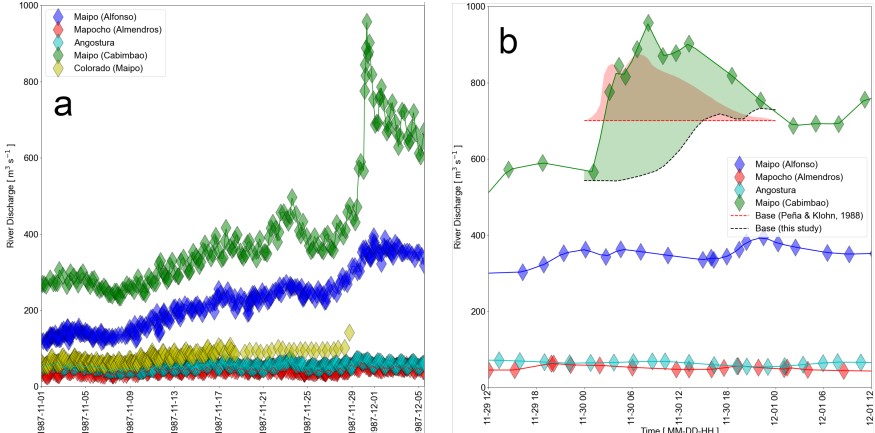

**Figure 6. Hydrographs from four gauge stations in the Maipo catchment.** For locations refer to Fig. 1. Shown is the record for the period from November 15 to December 5, 1987 (a) and for the event period (b). Also indicated are the baseline discharge (red dashed) assumed by na and Klohn (1988) as well as our baseline estimate (black dashed). The red shading indicates the Cabimbao hydrograph as reported by na and Klohn (1988) above a 700 m$^3$ s$^{-1}$ baseline. For this study, the flood volume - associated with the Parraguirre debris-flow event - is estimated from our baseline (green shading).

At the Cabimbao and San Alfonso gauging stations of Río Maipo, a gradual increase in discharge is observed after November 8 (Fig. 6a). At Maipo, values increase from 250 to about 400 m$^3$ s$^{-1}$ as typical discharge prior to the flood. From the Maipo hydrograph at Cabimbao, we confirm the Parraguirre flood event by a peak discharge on November 30 between 3:00 and 13:00. A clear maximum is visible at 8:03 of 956.8 m$^3$ s$^{-1}$. Afterwards discharge values settled at 650 m$^3$ s$^{-1}$. This step increase of ~250 m$^3$ s$^{-1}$ is independent of the debris flow and is largely explained by the headwaters of Río Maipo itself, where an

increase of ~150 m$^3$ s$^{-1}$ is visible at San Alfonso. As no liquid precipitation was recorded, melting of the extensive snow cover remains the main explanation. Considering a typical range of positive degree-day factors for snow between 3 and 7 mm w.e. °C$^{-1}$ day$^{-1}$, a warm spell of a few degrees could explain this increase in river discharge. At Alfonso the increase is recorded between November 28 and 30, a perfect match with the observed warm spell on 28 and 29.

The Cabimbao discharge record of Río Maipo is further utilised to infer a new volume estimate of the flood event associated with the Parraguirre debris flow (Fig. 6). The reason for this new estimate is that the earlier baseline discharge of 700 m$^3$ s$^{-1}$ appears too high, resulting in a low-biassed flood volume estimate of $7 \cdot 10^6$ m$^3$ (na and Klohn, 1988). The idea is to reconstruct a more reliable baseline discharge from the upstream discharge records at the Alfonso, Almendros and Angostura stations. First the travel times between these upstream stations and the Cambimbao reference are computed from a correlation analysis.

We find delays of 17, 19 and 6 hours for Alfonso, Almendros and Angostura, respectively. Subsequently, upstream time series are temporally aligned and discharge values are summed up. This bulk discharge was then scaled to match the Cambimbao reference on 29.11. and and 01.12, respectively. Scaling factors for these two days were linearly interpolated during the peak



discharge period on 30.11. The resultant background discharge (Fig. 6, dashed black line) was subtracted from the actually observed values at Cabimbao. A total flood volume of $16.0 \cdot 10^6$ m$^3$ is retrieved, more than twice the value previously reported (na and Klohn, 1988).

### 5.4 Simulations of the Parraguirre debris-flow propagation

*Experiment CAL1*

This first experiment served to calibrate the friction parameters in Area 1, such that the debris flow follows the observed impact area in the first 3 km. This primarily implies the 90°-turn southwards without overspilling to the west. We identified basal friction (DELTA1) as the controlling parameter and determined an ideal value of 20. Internal friction (PHI1) showed no significant impact and was therefore kept at the default value of 35.

*Experiment CAL2*

For this experiment, we deliberately separate the presentation into simulations with and without deposition.

Without deposition, the arrival time at the confluence Parraguirre and Colorado ranges between 5 and 8 min (10:38 - 10:41). This timing is much earlier than previously suspected (Table 1). Continuing downstream, an arrival time at La Paloma after 11:10 can only be guaranteed with a sufficiently large basal friction coefficient - i.e., DELTA2 > 3. Only considering these simulations as valid, the average arrival time at the confluence remains unchanged. Accepting both an earlier arrival time of 10:39 at the confluence and the observed arrival at La Paloma 11:20 implies average propagation speeds of the debris flow 4.3 m s$^{-1}$. This value is notably smaller than the average propagation speed further downstream in the valley section until Los Maitenes for the first wave (Table 1). This seems unlikely considering the decreasing valley slopes. This suggests that the Parraguirre debris flow might not have been a single-stage trigger-runout event. In the next paragraph, deposition is activated in our simulations to allow for temporal interruption of the flow propagation.

Activating deposition, all 90 members of the simulation ensemble show arrival at the confluence with Río Colorado and confirm the much earlier arrival times there. About half of the simulations still reach La Paloma, but much too early after 8 - 12 min (10:41 - 10:45). All of these show small basal friction. We discard them. The remaining 48 simulations all reach the confluence. At the confluence all simulations how heavy deposition in Río Colorado. For lower basal friction (DELTA2<5) and high values of relative entrainment (>70%), negligible deposition is simulated in Estero Parraguirre. We then assess the water volume that is mobilised at the time when the debris flow reaches the confluence. We assume that this volume has to explain the total flood volume of $16.0 \cdot 10^6$ m$^3$ observed at Cabimbao. Assuming an ample error margin of 66% the admissible ensemble size is reduced to 35 members. More decisive is the solid material volume of $38.1 \cdot 10^6$ m$^3$ exported from the Estero Parraguirre catchment. This volume is inferred from the observed elevation change. Assuming a permissible uncertainty of again ±66%, we retain 12 admissible ensemble members. In area 2, these show basal friction coefficients (DELTA2) of 3 or 4. Acceptable ranges of entrainment heights and solid entrainment fractions remain less constrained. After ranking the remaining





simulations according to the above fluid and solid volume constraints (sum of relative difference), the maximum entrainments height (HENTR in Area 2) is expected to fall between 6 and 8 metres, while the preferred solid fraction (RHENTR) is 70 or 80%.

*Experiment CAL3*

The remaining 8 experiments are primarily evaluated against the arrival time of the 2nd and main wave of the debris flow arriving around 12:37 at the Los Maitenes power plant. Decisive are the basal friction and the damming duration. The longer the damming time the later the arrival at Los Maitenes and the smaller the impact area (results not shown). These effects are

counterbalanced by reducing basal friction. To meet both targets - the timing and the impact area - a compromise is required. We deem DELTA2 = 3 and a damming time of 5 minutes as optimal (Table 4). The fluid and solid volume export from Estero Parraguirre are virtually independent of these two parameters and we can use these values to constrain the maximum entrainment height to 8 m. For a reasonable partitioning of solid and fluid volume, the relative solid entrainment height must exceed 70%.

In terms of performance, we opted for three metrics: the factor of conservativeness (FoC), the critical success index (CSI) and the distance to perfect classification (D2PC). The latter two quantify the degree of overlay between the simulated and the observed impact area. FoC however also measures if the predictions over- or underestimate the observed impact area. In our case, all measures remain far from their optimum. Values indicate at best moderate performance (cf., Mergili et al., 2018a, b).

FoC indicates that the impact area is significantly overestimated by a factor 2 to 3. Unfortunately, these performance metrics cannot serve for further calibration. Best values also conflict with the arrival time at Los Maitenes.

In terms of spatio-temporal progression of the simulated debris-flow (Fig. 7), we re-confirm that the arrival time agrees fairly well with the few available observations (Tables 1, 4). The spatial flow-height pattern shows that the main wave front of the debris flow exceeds the observed impact area at all times. The resultant maximum flow height map (Fig. 8) was used to com-

pute the above performance metrics (Table 4) - confirming the overestimation of the actual debris-flow extent. After the first wave passes, the debris flow is rather well confined and closely follows the deepest valley section. Downstream of El Alfalfal, the valley floor is characterised by wide and flat plains. There the observed impact area is confined to the narrow riverbed. The simulated debris flow occupies, however, most of the valley floor.

**6   DISCUSSION**

**6.1   Meteorological pre-conditioning**

*Climate reanalysis*

The climate reanalysis points at a prevailing high-pressure in the Eastern Pacific associated with southerly winds along the coast at the end of November (25-29). Near central Chile, the synoptic circulation indicates easterly flow across the Andes



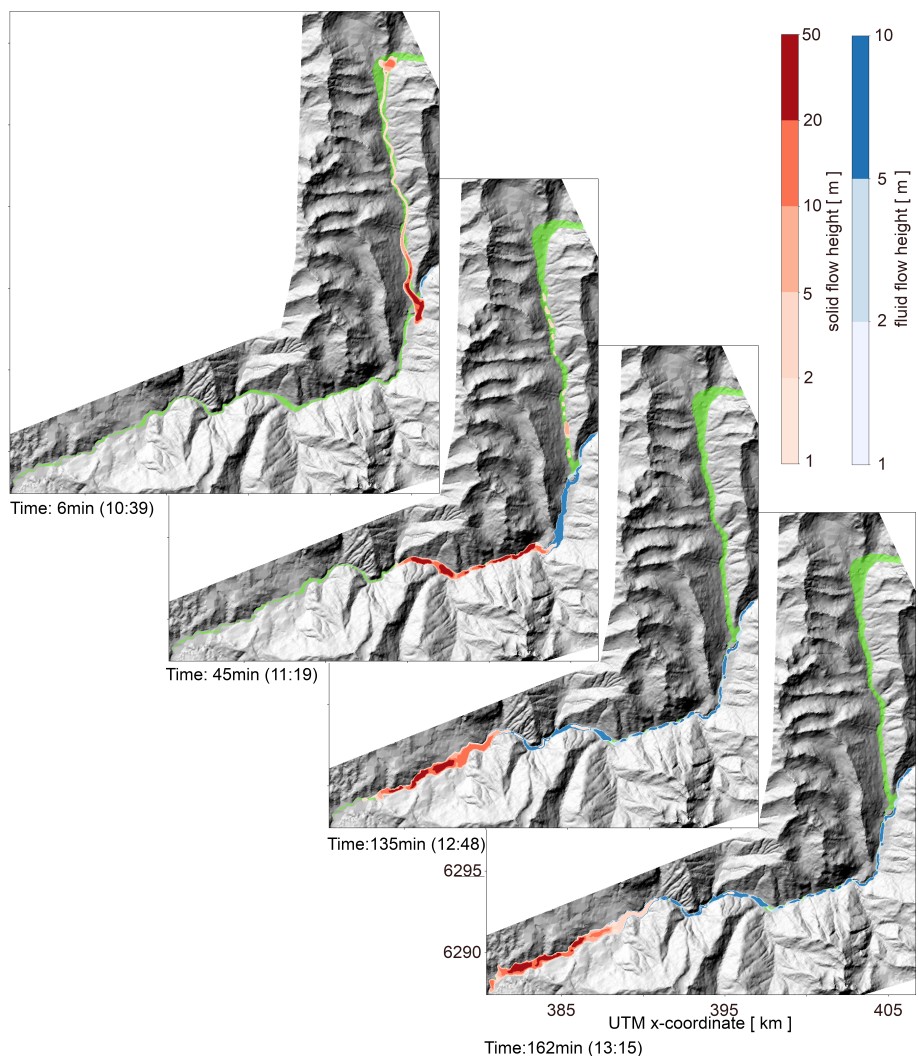

**Figure 7. Fluid and solid flow heights of Parraguirre debris flow.** Shown are results for experiment 3 (CAL3) using PHI1=35°, PHI2=6°, DELTA1=20°, DELTA2=3°, HENTR=8m and RHENTR=70%. Panels show moments when the debris flow reaches prominent sites: confluence with Río Colorado, La Paloma camp site and Los Maitenes. The last panel shows the final flow height after 10'000 simulation seconds. Red and blue colours indicate solid and fluid flow heights, respectively. The green shaded area represents the observed impact area. Background: Hillshade historic DEM 1955.

resulting in Foehn-like conditions on the western slopes, mainly at higher elevation. In the same year, the reanalysis, consistently with snow surveys, suggests anomalous high snow cover in 1987. Moreover, a long-lasting and strong warm spell was present during the avalanche previous days. We therefore forward the hypothesis that the Parraguirre ice-rock avalanche was a meteorological compound event (e.g., Zscheischler et al., 2020). 1987 stands out as the only year on record in which anomalous snow cover coincided with exceptional warm periods in spring. However, geo-tectonical and mechanical aspects



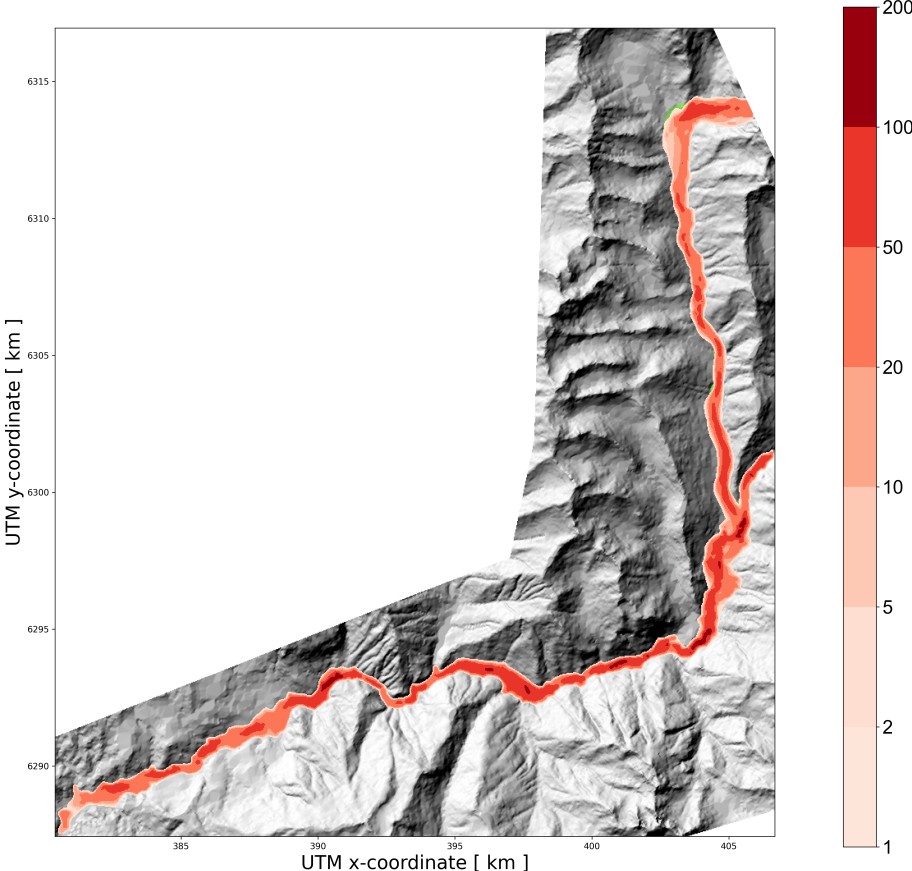

**Figure 8. Maximum flow height of Parraguirre debris flow.** Shown are results for experiment 3 (CAL3) using PHI1=35°, PHI2=6°, DELTA1=20°, DELTA2=3°, HENTR=8m and RHENTR=70% for experiment 3 (CAL3). Background: Hillshade historic DEM 1955.

are also relevant (Sepúlveda et al., 2023).

*Station data*

Snow height measurements in the Maipo catchment support extreme values in 1987 being second place in the 50-year record. This confirms the meteorological pre-conditioning. Yet photographs acquired 2 days after the Parraguirre event show that a
solid snow cover was only present on south-facing slopes or at elevation above 3 500 m a.s.l. (Fig. 2e). Concerning anomalous temperatures, the weather station shows rather cold weeks until mid of November. Then a steep increase is visible with warm conditions prevailing until the end of the month - with a clear peak on November 28 and 29. These station records therefore confirm the hypothesis for a meteorological compound event.



*Consistency with hydrological record*

The observed temperature increase after mid November perfectly matches a first increase in river discharge of Río Maipo (Fig. 6). Also the two-day warming peak is well imprinted in the gauge records. As a result, the November discharge value in 1987 for Río Colorado was twice as high as on average (Table 3). As no rainfall was recorded, these elevated values likely resulted from extensive snow melt. Assuming that continuous snow cover is limited to areas above 3 000 m a.s.l., an addi-
tional 40 $m^3$ $s^{-1}$ discharge can only be sustained by an extra snow melt of 5-10 mm per day (20-30 cm per month) above the annual average. This extra snow-melt requires positive anomalies in monthly mean temperatures between 0.5 and 1°C. The weather station at El Yeso Embalse recorded a monthly temperature anomaly of about 1.2°C in 1987 against the long-term mean for November (1970-1999). From these considerations, elevated river discharge can fully be explained by snow melt. We are therefore convinced that the sub-surface of mountain flanks and valley floors was highly water saturated prior to the
Parraguirre debris-flow event. This is especially true for the two very warm days preceding this event (Suppl. Fig. S3a). Both the gauge measurements and the meteorological record draw a consistent picture of anomalous meteorological conditions in 1987 and further substantiate the hypothesis of the Parraguirre rock avalanche to be a meteorological compound event.

## 6.2   Experimental Setup

*Trigger volume*

We present the first estimate of the trigger volume based on DEM differencing with a volume of $17 \cdot 10^6$ $m^3$. Previous estimates of about $6 \cdot 10^6$ $m^3$ are significantly smaller (Casassa and Marangunic, 1993; Hauser, 2002)). Yet these were loosely inferred from visual inspection during field visits. Hauser (2002) describes the initial extent of the destabilised rock slab as a rectangle with a length of 1'000 m, a width of 500 m and a mean thickness of 20 m. This description already indicates a larger initial
volume of about $10 \cdot 10^6$ $m^3$. Naranjo et al. (2001) estimated $14 \cdot 10^6$ $m^3$ from an extent 700m x 400m x 50m. From DEM differencing, we confirm the rectangular shape of the trigger area but we find a slightly larger extent than both field inspections: 1'100 m long and 600 m across. The average thickness is 26 m. We are convinced that the larger horizontal extent is well constrained from remote sensing (Sect. 4.1). The absolute slab thickness, as inferred from DEM differencing, is certainly less reliable. Part of the uncertainty stems from difficulties in co-registration of the aerial photographs and from the fact that the
reference DEM dates back to 1955.

*Calibration targets*

In CAL1, friction coefficients in area 1 are adjusted such that the debris flow does not overtop the glacier in the west and does turn southward. We consider this a robust target because it is confirmed independently from the impact area as well as from
super-elevation observations (Casassa and Marangunic, 1993). In CAL2, the late arrival time at La Paloma serves to discard about half of the simulations. As all these simulations show much too early arrival at this site, the precise timing is not critical for the calibration. We are therefore confident that our simulations point at a multi-stage debris-flow event in Parraguirre. The final calibration also considers the fluid content of the debris flow on arrival at the confluence with Río Colorado as well as the



solid mass export from Estero Parraguirre. The former shows a value of $16 \cdot 10^6$ m$^3$ - being much higher than previously thought

(Casassa and Marangunic, 1993; Hauser, 2002). Yet this flood volume is rather reliable because our estimate is substantiated by actual river-discharge observations. The more decisive calibration target is the solid mass export. We admit considerable uncertainties in our estimate of $38.1 \cdot 10^6$ m$^3$, which is reflected in the large admissible calibration range of $\pm 66\%$. While relative and max. absolute entrainment heights exert control on the partitioning between fluid and solid volume in these simulations, this calibration step also favours a certain range of friction coefficient. This friction range is confirmed by CAL3 to assure ap-

propriate arrival time around 12:37 at Los Maitenes (Table 4). Without damming, additional experiments with identical friction coefficients show no arrival before 13:20 at Los Maitenes (results not shown). The water pulse from the dam breach of Río Colorado – though small in volume – is therefore essential for a realistic debris-flow propagation.

*Experimental design*

An asset of our simulations is the simplicity of the setup. We only require a partitioning of the impact area into two sub-regions with distinct friction coefficients. These consistently explain both fluid and solid mass transfer as well as the observed arrival times at two locations. Nonetheless, we admit that our experimental setup remains highly idealistic - specifically w.r.t. three aspects: fluid entrainment, temporal damming, deposition and resolution.

First, we mimic the immense water uptake by an entrainment map with a fixed maximum height and a prescribed relative fluid portion. The best simulation suggests a maximum entrainment height of 8 m including about 2 metres of water ($\rho_{\text{water}} = 1000$ kg m$^3$). This could suggest massive snow cover along the valley floor of Estero Parraguirre. Valenzuela and Varela (1991) reported on a 5 - 10 m snow cover at the time ($\rho_{\text{snow}} = 300 - 500$ kg m$^3$) - an adequate value to explain such water columns. Unfortunately, the DGA photographs along the Parraguirre valley from just two days after the rock avalanche disprove ex-

tensive snow cover. Even the 1-m snow-cover estimate, that was forwarded by Casassa and Marangunic (1993) and judged insufficient ($\sim 20\%$) to explain the total mobilised water volume, seems unlikely. Moreover, the 30-min damming of Río Colorado can only explain less than 1% of this volume. Assuming no further significant contribution to the total flood volume downvalley of the confluence with Río Colorado, three other possible sources of water input remain: entrainment of other water bodies, of glacier ice, or of saturated soil along the valley floor. First, no persistent water bodies are visible on optical

imagery (Landsat 5) from February to November in Estero Parraguirre or in the pro-glacial area. En- or subglacial water bodies cannot be excluded but a substantial size would have been required. Water heights of 10 or 50m would have been required beneath the entire glacierised area or the removed portion of Glacíar 24, respectively. Considering the topography, both scenarios seem unlikely. Second, Glaciar 24 lost about 0.3 km2 of its area, which is well imprinted in the elevation change map showing an average surface lowering of 32 m. This lowering translates into a low-end estimate for released water volume of

$8-9 \cdot 10^6$ m$^3$. The associated melt energy is however substantial and it seems unlikely that the melting was completed within Estero Parraguirre. The reason is that the total displaced solid material was only 2.5 times larger than the total flood volume. Heat transfer within the solid-fluid-frozen mixture sufficient for melting would imply the solid material should have been significantly warmer (>10°C). Third, the last source of water is saturated soil along the valley bottom. Casassa and Marangunic



(1993) estimated a total fluid volume of about $0.6 \cdot 10^6$ m$^3$. Considering the warm atmospheric conditions and the elevated rates of snow melting, the soil and sub-surface must have been saturated with water. Global mapping of available soil water indicates 10-20% potential plant-available soil water in the capital region of Santiago (e.g., Zhang et al., 2018; Gupta et al., 2023), with field capacities being slightly higher. Considering the total solid material export of $38.1 \cdot 10^6$ m$^3$, we would expect at most $8 \cdot 10^6$ m$^3$ additional water. In summary, the $16 \cdot 10^6$ m$^3$ total flood volume can only be explained as a combination of highly saturated soils in the Estero Parraguirre valley and the entrainment of glacier ice. Ice entrainment is impractical to directly incorporate into the model as the removed ice volume is unknown and the representation of gradual melting during debris-flow propagation exceeds the current capabilities of r.avaflow. In our setup, we therefore decided to mimic fluid uptake (from ice melting and soil entrainment) as a gradual process during flow propagation.

The second idealised aspect concerns the hypothesis for damming of Río Colorado. Hauser (2002) used the argument of the late arrival at La Paloma to substantiate his hypothesis. Here, we can use a model that describes this damming as a temporal water release (i.e. input hydrograph). The damming is defined by three parameters: the released water volume, the timing and the duration. The released water volume was motivated above from the damming time and the discharge estimate of Río Colorado based on a scaling argument (see Sect. 5.3). Both seem moderately well constrained. As the model suggests much faster propagation speeds along Estero Parraguirre, we forward a damming time of 30 min, rather than the 15 - 20 min suggested by Hauser (2002). The duration of the water release is set to 5 minutes. This duration is a compromise to reproduce the arrival time at Los Maitenes as well as the impact area. Without damming, the debris flow barely reaches El Alfalfal around 12:37. At that time it was already observed 7-8 km downstream at Los Maitenes - impossible to reach in time with typical propagation speeds of about 4-5 km hr$^{-1}$ (not shown). The forwarded damming time and breached water volume should serve as a first orientation to constrain the multi-stage debris-flow event, which we deem necessary for the late arrival at La Paloma camp site.

The third aspect relates to the idealistic decision to deactivate deposition (i.e., a stopping criterion) for the simulation in CAL3. This was necessary to ensure arrival at Los Maitenes. Admittedly, the details of the stopping criterion could be adjusted as remedy. However, we did not have a specific interest in the final deposition along the narrow Río Colorado for validation, as values there appear questionable (also see next paragraph).

The fourth and last idealistic aspect relates to the model resolution of 30 m. This resolution might well be adequate to describe the first phase of the debris flow until deposition and damming of Río Colorado. There, the impact area is well resolved (Fig. 8). Yet further downstream, it is largely overestimated. The reason is that the impact area closely follows the river and its direct vicinity. In most of the river sections with a flat and wide valley floor, this area is, however, entirely occupied by the debris flow. Along Estero Parraguirre, a resolution increase from 60 m to 30 m was key to better reproduce the impact area (results not shown). Further downstream the improvement is less prominent.





## 6.3 Simulation cascade

The intention of our experimental design is not to present a best simulation that insinuates to re-present the most realistic
scenario of the Parraguirre debris flow. Our modelling efforts remain idealistic and are limited both by observational evidence
and the DEM resolution. The main motivation is to substantiate our main conclusion on the event cascade from physical-based
modelling. Our experimental setup provides evidence to answer the following questions:

1. Is the rather large trigger volume plausible?

   Our DEM differencing suggested a trigger volume more than twice as large as previously assumed (17 as against $6 \cdot 10^6$
   m³). CAL1 suggests that the simulated debris flow can be restrained within the observed impact area using default
   friction parameters. From a modelling perspective, our large estimate for the trigger volume is reasonable. Significantly
   smaller trigger volumes would involve higher fluid and solid entrainment to explain the observations. These are already
   barely reconcilable with our high-end estimate of the trigger volume (cf. Sect. 6.2).

2. Where lies the source area for the large flood volume?

   Another high-end estimate is forwarded for the fluid flood volume from our analysis of the river gauge stations. Despite
   the idealistic character of the entrainment process in our simulations (Sect. 4.2), a realistic amount of water and solid
   material is exported from Estero Parraguirre. Primary water sources are glacier ice, soil water and snow cover. Water
   input from glaciers is mandatory to keep assumed soil-water saturation values in a realistic range. Our simulations
   therefore corroborate that water availability in the Estero Parraguirre catchment did suffice to explain the total monitored
   flood volume.

3. Was the Parraguirre debris flow a multi-stage event?

   Despite the first speculations from Hauser (2002), more robust evidence is yet missing for the damming hypothesis of
   Río Colorado. Our simulations offer a first remedy hinging upon the two available observations of arrival times (La
   Paloma, Los Maitenes). Assuming that these observations are dependable, our simulations show that on-time arrival at
   La Paloma imply an important delay in arrival at Los Maitenes. If arrival is reproduced at Los Maitenes, the campsite at
   La Paloma should have been reached more than half an hour earlier. Temporal cessation and re-activation of the debris
   flow provides a simple and convincing explanation. Respective simulations can reproduce run-out distances and arrival
   times. A temporal damming of the deeply incised Río Colorado is further supported by observations on the immense
   solid volume transferred downvalley and by mean deposition heights of 3 m on the confluence plain between Estero
   Parraguirre and Río Colorado (Table 1, Casassa and Marangunic, 1993).

4. Can observations on impact area and maximum flow heights be explained?

   Both quantities are overestimated in our simulations. The unknown release time after damming exerts primary control
   on these quantities. Without damming, maximum flow heights of 17 and 18 m (Naranjo et al., 2001) and superelevation
   marks of 40 m (Casassa and Marangunic, 1993) can well be reproduced along Río Colorado upstream of La Paloma



(Table 1). Yet the impact area remains overestimated, particularly along the incised valley of Río Colorado. In this
        regard the DEM resolution appears as main limitation.

Finally, we see several leads to further improve our best simulations in terms of reproducing timing and impact area. We
suggest a refined partitioning of the region with regard to friction and entrainment maps. Moreover, these final simulations did
not include deposition. Therefore elevation changes further down-valley could not directly be considered for calibration and
validation.

### 6.4   Mass movement and future challenges across the Semiarid Andes

The Parraguirre ice-rock avalanche was one of the most catastrophic events in the semiarid region of Chile. While several other
landslides have occurred in the area (Moreiras et al., 2021), none exhibited a magnitude comparable to that of the Parraguirre
event. A smaller debris flow was detected in the La Difunta Correa catchment (30°S), near the Aguas Negras mountain pass.
There, the seasonal snow and shallow ice within the active layer played a pivotal role (Vergara Dal Pont et al., 2020). The
majority of other documented events in the region are linked to heavy precipitation during positive phases of El Niño–Southern
Oscillation (ENSO) or the passage of prominent weather fronts.

Mass movements can have important consequences for the mountain environment and landscape with direct impact on local
communities. In a warming climate, the frequency and intensity of extreme weather events are likely to increase. The rising
occurrence of heatwaves and warm spells is expected to alter mountain environments with regard to slope stability as well as
rock and debris mobilisation. For instance, mountain permafrost, one of the essential climate variables, has shown a marked
decline due to global atmospheric warming (Chen et al., 2021). Occurring in a mountain permafrost region (Gruber, 2012),
the Parraguirre ice-rock avalanche provides valuable information on the severe consequences of climate change in the sucep-
tible areas. The degradation of mountain permafrost is becoming a growing concern worldwide, as it directly affects local
populations through increased hazard susceptibility (Chen et al., 2021).

### 7   CONCLUSIONS

The main motivation for this holistic review of the 1987 Parraguirre rock avalanche on November 29 is threefold. First, clima-
tological and hydrological stations data as well as atmospheric reanalysis products have become publicly available. Second,
aerial images from within two weeks after the event shed new light on the debris-flow extent. Third, state-of-the-art modelling
tools can serve to constrain the evolution and propagation of past multi-phase gravitational mass movements. Many of these
records and methods were not readily accessible for existing studies (Ugarte, 1988; Valenzuela and Varela, 1991; Casassa and
Marangunic, 1993; Naranjo et al., 2001; Hauser, 2002). The combined analysis is novel and allows us to better re-draw the
debris-flow event from its initiation at $4\,200$ m a.s.l. to its termination at $1\,100$ m a.s.l. – about 50 km away from the source.



*Meteorological pre-conditioning*

In-situ station data and climate reanalysis independently indicate the extraordinary atmospheric conditions in 1987 with significantly elevated snow depths and unusually long and pronounced warm spells. Weather stations show a 2-day warm spell on November 28 and 29 with average temperatures of 5°C above the long-term average. The magnitude of monthly temperature anomalies together with the prominent snow cover suggest an additional 40 m$^3$ s$^{-1}$ discharge of Río Colorado. This is consistent with hydrological measurements in November 1987. Independent of the ultimate trigger mechanism, the Parraguirre ice-rock avalanche and its development into a devastating debris flow has to be considered a meteorological compound event because soils and snow cover were highly water saturated at the time.

*Debris-flow extent and volume* We present the first high-resolution map of the impact area based on areal imagery. The areal extent amounts to 12.1 km$^2$ and the run-out distance reaches 50 km. Moreover, we forward a significant upward correction of the trigger volume to 17.0·10$^6$ m$^3$ together with the first estimate of solid material volume of 38.1·10$^6$ m$^3$ being exported from Esterro Parraguirre. These estimates rely on geodetic methods using aerial imagery. Finally, the total flood volume of 16.0·10$^6$ m$^3$ (fluid portion) was recalculated on the basis of the gauge station in Cabimbao. The total debris-flow volume, including fluid and solid material, accounts to 54.1·10$^6$ m$^3$ - also being an unprecedented estimate.

*Multi-stage debris-flow propagation*

Our model simulations strongly suggest that the event history has to be rewritten. Two observations on arrival times suggest that the debris-flow must have paused upstream of La Paloma resulting in a temporary damming of Río Colorado. This dam breach hypothesis reconciles the arrival-time observations with our simulations. This hypothesis was first forwarded by Hauser (2002) and is finally substantiated by state-of-the-art modelling techniques. The impact area and individual observations of maximum flow height are moderately well reproduced - possibly related to a limited DEM resolution.

*Code availability.* The r.avaflow code, including a detailed manual, is available for download at https://www.avaflow.org/ (Mergili and Pudasaini, 2024).

*Data availability.* Analysis and simulation data can be acquired on request from the corresponding author. Input data is either publicly available or under copyright. In the following, we point the interested reader to the respective data holders and pertinent repositories.

Publicly-available data sets: ERA5 data are available from the Copernicus Climate Data Store (CDS). Weather-station, gauge-station and snow height data are freely distributed via open repositories administered by the 'Dirección Meteorológica de Chile', being integrated in the 'Dirección General De Aeronáutica Civil' (DGAC), and the 'Dirección General de Aguas' (DGA) as part of the 'Ministerio de Obras Públicas' (MOP) of Chile. Three overview data portals DGA Red Hidrométrica, DGA MAPAS and the Catastro de Estaciones del Sistema



SACLIM give information on station locations and operational details. Specific variables can be retrieved from the DGASAT and BNA data sets.

Proprietary data sets: Historical elevation models are administered by and can be requested or purchased from the Geographical Military Institute (IGM) of Chile. The aerial imagery used in this study were provided on request from the 'Servicio Aerofotogramétrico' (SAF) of the Chilean Air Force.

*Author contributions.* D.F.-B. had the idea to overhaul the history of the Parraguirre event and initiated the acquisition of the historic maps.
595 J.J.F and D.F.-B. designed the study together. Aerial imagery were analysed and processed by D.F.-B. to infer the impact area and elevation changes. T.B. and D.F.-B. conducted the first experiments with r.avaflow, which were subsequently refined by J.J.F: resolution increase, domain extension, calibration strategy, damming experiments. The analysis of climatic conditions was led by L.S. and supported by D.F.-B., T.B. and J.J.F. Hydrological observations were compiled and assessed by J.J.F. S.M and H.P. supported the overall analysis with their long-track expertise in mountain hydrology and mountain hazards of the Capital region of Chile. All authors contributed to the interpretation of
600 the results and to the writing of the manuscript, both under the coordination of J.J.F and D.F.-B.

*Competing interests.* The authors declare no competing interests.

*Disclaimer.* Neither the European Commission nor ECMWF is responsible for any use that may be made of the Copernicus information or data it contains.

*Acknowledgements.* The main author J.J.F. received primary funding from the European Union's Horizon 2020 research and innovation
programme via the European Research Council (ERC) as a Starting Grant (FRAGILE project) under grant agreement No 948290. D.F.-B. was funded by the MAGIC project financed by the German Research Foundation (DFG) within the MAGIC and ITERATE projects (FU1032/5-1, BR2105/28-1, FU1032/12-1) as well as the programa Postdoctorado, VRID Universidad de Concepción, ANID Subvención a la instalación a la academia 2022 (PAI85220007), and Anillo ACT210080 and Fondecyt 3230146. L.S. and D.F. acknowledge the Anillos de Investigación en Areas Temáticas, Cold-Blooded: Drivers of Climate Change Refugia for Glaciers and Streamflow Responses, ACT-210080. L.S. also
acknowledges the support of the supercomputing infrastructure of the NLHPC (CCSS210001) to develop this research, Powered@NLHPC. Moreover, we greatly acknowledge the Geographical Military Institute (IGM) of Chile and the 'Servicio Aerofotogramétrico' for historical maps and aerial photographs, respectively. The results contain modified Copernicus Climate Change Service (C3S) information. Finally, the authors gratefully acknowledge the scientific support and HPC resources provided by the Erlangen National High Performance Computing Center (NHR@FAU) of the Friedrich-Alexander-Universität Erlangen-Nürnberg (FAU). The hardware is funded by the German Research
Foundation (DFG).



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



**Table 1. Overview Table of event history.** Distance from the trigger area, elevation and slope were re-determined (normal font) and are compared to values reported in previous studies (italic). For consistency, we mostly rely on our estimates. For timing, velocities, deposition and flow heights, we entirely rely on previous reports. Inconsistent or less reliable timing values are indicated in italic. Subscripts for timing indicate elapsed minutes after 10:33. Superscripts point to references.

| Location | Distance *from source* [km] | Elevation [m a.s.l.] | Surface Slope *mean to previous location* [deg] | Date/Time *1st wave (2nd/3rd)* | Velocity Estimates [m s⁻¹] | Mean Velocity *w.r.t. previous location/source 1st wave (2nd)* [m s⁻¹] | Deposition Height *local/mean* [m] | Max. Flow Height [m] |
|---|---|---|---|---|---|---|---|---|
| **29 . 11 . 1987** | | | | | | | | |
| Source Area/ Valley Head | 0-1 | 3800 − 4600, 3400 − 4350[1], 4000 − 4500[2] | 38 (20-80), 70-75 ( - )[1] | $10{:}33^{6}_{0}$ (seismic tremor of impact) | | | | |
| Upper Valley / Glacier / Valley Turn | 1-2.5 | 3300 − 3600 | - (5-20) | | > 31 (energy head)[1], 24 (super-elevation)[2] | | | 50[1,2] |
| Estero Parraguirre | 2.5-17 | 2100 − 3300 | 4.9 ( - ), 4.5 ( - )[6] | | 15 (volume flux argument)[4] | | | |
| Confluence Parraguirre / Colorado | 17, 17[1,2] | 2050 − 2150 | - (0-1) | $10{:}52_{19}$ (assuming 15 m s⁻¹) | ~ 10 (eye-witness)[1,2] | 14.9 / 14.9 | - / 3[2] | |
| Río Colorado (Parraguirre to La Paloma) | 17-27.5 | 1630 − 2100 | 2.1 ( - ) | | | | | mean 4[2], max. 4[2], 18 (Tambillo)[3], 17 (Espinoza)[3] |
| La Paloma Camp | 27.5 | 1630[1] | - (0-10) | $11{:}20^{5}_{47}$ | | 6.3 / 9.8 | 0.6[1] / -, - / 3[2] | |
| Confluence Olivares / Colorado | 31.5 | 1560 | 1.0 ( - ) | - ( - / $16{:}20_{347}$)[2,7] | | | | 50[2], 10 (Ranchos)[3] |
| El Alfalfal (power house[1]) | 38.5, 41[1], 40[2,3] | 1360, 1360[2], 1200[1] | 1.6 | - ($12{:}37_{124}$ / - )[2], $12{:}14$ ($12{:}37_{124}$ / - )[3] | | - (2.4) / - (5.2) | 4[1] / -, 2[2] / -, 1.5[3] / - (river bed), 0.6[3] / - (lateral) | 8[3] |
| Los Maitenes (hydroelectrical plant[1,2]) | 46, 41[1,2] | 1260, 1150[1] | 0.8 | $12{:}14_{101}$ ($12{:}37_{124}$ / $16{:}20_{347}$)[1], $12{:}14_{101}$ ( - / - )[2] | | 5.0 (4.0) / 7.6 (6.2) | 2.5-2.7[1] / - | 30-35[1] |
| **30 . 11 . 1987** | | | | | | | | |
| Cabimbao (Maipo gauge) | 223[1], 220[2] | 0 | 0.4 | $03{:}20^{2,7}_{200}$ ($07{:}00^{1}_{1227}$ / - ) | | 3.2 (2.6) / 3.7 (3.0) | | |

[1] Hauser (2002)  [2] Casassa and Maranguníc (1993)  [3] Naranjo et al. (2001)  [4] na and Klohn (1988)  [5] Valenzuela and Varela (1991)  [6] Eisenberg and Pardo (1988)  [7] Ugarte (1988)





**Table 2. Experimental Design.**

| calibration experiments | description & calibration targets | parameter ranges |
|---|---|---|
| CAL1 calibration | Impact area follows 90° turn into Estero Parraguirre <br><br> *(full sim. time: 200 s/3.3min/10:36:20)* | Area 1 <br> PHI1= 15, 25, 35 <br> DELTA1= 10, 15, 20, 25, 30 |
| CAL2 calibration | - reach confluence Colorado <br> - arrival time La Paloma <br> - flood volume from gauge (Cabimbao) <br> - solid mass export from Estero Parraguirre <br><br> *(full. sim. time: 2900 s/48.3min/11:21:20)* | Area 2 <br> PHI2=2, 4, 6, 8, 10 <br> DELTA2=PHI/2 <br> HENTR2=2, 4, 6, 8 <br> RHENTR2=0.5, 0.6, 0.7, 0.8 <br> Deposition= 0, 1 |
| CAL3 | - flood volume from gauge (Cabimbao) <br> - solid mass export from Estero Parraguirre <br> - arrival time Los Maitenes <br><br> *(full. sim. time: 10'000 s/166.6min/13:19:40)* | Hydrograph Colorado <br> Start: 11:10 *(37min)* <br> Duration= 1, 2, 5, 10 min <br> Total hydrograph <br> fluid volume = 51'000 m$^3$ |

**Table 3. Typical river discharge regime for the Maipo catchment.** For the multi-annual discharge estimates, standard deviations are indicated together with the respective time period (subscripts). 1987 values are highlighted in bold if they exceed these standard deviations.

| gauge station | catchment area <br> [ km$^2$ ] | November discharge multi-annual average <br> [ m$^3$s$^{-1}$ ] | November discharge 1987 <br> [ m$^3$s$^{-1}$ ] |
|---|---|---|---|
| Maipo (Cabimbao) | 15'040 | $145.8 \pm 90.0_{1940-1990}$ | **355.7** |
| Mapocho (Almendros) | 620 | $13.7 \pm 9.3_{1948-1990}$ | **41.3** |
| Angostura | 1'394 | $26.7 \pm 12.9_{1981-1990}$ | **54.0** |
| Maipo (Alfonso) | 2'850 | $105.9 \pm 35.2_{1940-1990}$ | **206.4** |
| Colorado (antes Maipo) | 1'713 | $38.9 \pm 15.7_{1940-1990}$ | 81.9 (data until 28.11.) |
| Colorado (antes Olivares) | 834 | $22.4 \pm 6.7_{1977-1990}$ | 22.3 (data until 12.11.) |
| Olivares | 531 | $13.2 \pm 4.4_{1977-1990}$ | 16.0 (data until 12.11.) |



**Table 4. Results and performance from the damming experiments.** Simulations are based on CAL3 using the calibration results from CAL2. The best two options per performance criteria are marked in bold and larger font size. For the performance metrics (Formetta et al., 2016; Mergili et al., 2017), we define areas with observed mass flow impact as observed positives (OP) and the ones without observed mass-flow impact as observed negatives (ON). Predicted positives (PP) and negatives (PN) are areas with and without simulated mass-flow impact, respectively. From these quantities, we infer true positive (TP), true negative (TN), false positive (FP) and false negative (FN) predictions.

| 5-min damming | Parameter combinations | | | | | | | |
|---|---|---|---|---|---|---|---|---|
| basal friction (DELTA2) [ ° ] | 4 | 4 | 4 | 4 | 3 | **3** | 3 | 3 |
| max. entrainment height (HENTR) [ m ] | 8 | 8 | 6 | 6 | 8 | **8** | 6 | 6 |
| rel. solid entr. height (RHENTR) [ % ] | 80 | 70 | 80 | 70 | 80 | **70** | 80 | 70 |
| exported fluid volume *target: 16.0* [ $10^6 m^3$ ] | 10.8 | **15.2** | 9.5 | 13.1 | 11.9 | **16.5** | 10.2 | 13.9 |
| exported solid volume *target: 38.1* [ $10^6 m^3$ ] | **37.2** | 32.5 | 30.0 | 26.2 | **37.5** | 32.8 | 30.3 | 26.5 |
| arrival at Los Maitenes *target: 12:37* (±3min) | >13:20 | >13:20 | >13:20 | >13:20 | 13:06 | **12:48** | 13:00 | **12:42** |
| factor of conservativeness (FoC) *(optimal 1)* FoC = PP/OP = (TP+FP)/(TP+FN) | **2.146** | 2.250 | **2.195** | 2.294 | 2.655 | 2.754 | 2.684 | 2.770 |
| critical success index (CSI) *(optimal 1)* CSI = TP / (TP+FP+FN) | **0.422** | 0.411 | **0.415** | 0.405 | 0.367 | 0.357 | 0.365 | 0.355 |
| distance to perfect classification (D2PC) *(optimal 0)* D2PC = sqrt((1-rTP)**2+rFP**2) rTP = TP/OP rFP = FP/ON | **0.251** | 0.266 | **0.259** | 0.274 | 0.335 | 0.354 | 0.340 | 0.357 |