# Peer review of "The Parraguirre ice-rock avalanche 1987, semi-arid Andes, Chile -A holistic revision"

_EGUsphere, 2024_

## Referee Comment (RC3)

**Reviewer's comments**

**Ref:** egusphere-2024-3103
**Title:** *The Parraguirre ice-rock avalanche 1987, semi-arid Andes, Chile -A holistic revision*

**General Comments:**
In this study, the authors simulated the 1987 Parraguirre Debris flow by using existing records with hydro-climatological data, remote sensing observations, and process-based modeling. For simulation, the *r.avaflow* (open access software) has been used. Their simulated model has been validated with existing results of previous studies and field observations. The research is interesting and important to predict the future ice-rock avalanche for global warming due to climate change, but not a novel one.

**Specific Comments:**
1. There are grammatical errors and typos (such as line 2 (Abstract): instead of 'is' here will be 'are'). The authors should go through a thorough editing of the manuscript to correct grammatical errors and improve the readability of the paper, using a professional English editor as appropriate.

2. The 30-m resolution Digital Elevation Models (DEM) generated from a 1:50,000 scale topographic map and aerial photographs have been used in this study. The volume of debris and water calculations of the 1987 avalanche using re-registering (resampling) of a 30-m resolution DEM is not accurate and precise. Therefore, the volume of debris and water is much higher than in the previous studies. Describe this limitation in the discussion section.

3. The weather stations are sparsely distributed in the study areas. The weather parameters, such as temperature, precipitation taken from 40 to 50 km away from the areas where the volume of melt ice was calculated. The temperature within 40/50 km may vary several degrees. Therefore, this limitation should be explained in the discussion section.

4. This type of limitation is also observed in the hydrological data; the limitation of the hydrological data (discharge) should be explained in the discussion section.

5. *The caption of the figure should be numbered first, and then write the text of the caption, such as (a) Southern South America and (b) inset of the Metropolitan*

6. *Technically, I think the method described in the manuscript is sound, but not innovative.*

**Decision: Major revision**

---

## Author Comment (AC2)

Point-by-Point Reply to Reviewer Reports on the manuscript entitled

**The Parraguirre ice-rock avalanche 1987, semi-arid Andes, Chile**
-
**A holistic revision**

by J.J. Fürst, D. Farías-Barahona et al.

**First of all, we want to thank all reviewers for the critical and useful comments on the manuscript. It was with great pleasure that we read their general appreciation of our work and the unanimous recommendation for publication. In the following, all review comments are repeated and addressed. Our responses are indented. Answers are given in blue and actions in orange. *Italic font* is enquoted and used to present newly added or modified text passages in direct answer to a specific comment.**

We are very delighted about the fact that all three reviews are positive and supportive for publication in 'Natural Hazards & Earth System Sciences' (NHESS). Most comments have an editorial character. We tried to accomodate them to the best of our knowledge. If reviewer comments disagree, a compromise is suggested. Furthermore, reviewer #3 requested additional discussion of limitations of the employed methods (DEM differencing, climate data scarcety, ...). We accounted for these. In summary, we hope that the editor can continue to consider our manuscript for publication in NHESS.

**REFEREE #3**

Reviewer's comments
Ref: manuscript egusphere-2024-3103
Title: The Parraguirre ice-rock avalanche 1987, semi-arid Andes, Chile -A holistic revision

General Comments:

In this study, the authors simulated the 1987 Parraguirre Debris flow by using existing records with hydro-climatological data, remote sensing observations, and process-based modeling. For simulation, the r.avaflow (open access software) has been used. Their simulated model has been validated with existing results of previous studies and field observations. The research is interesting and important to predict the future ice-rock avalanche for global warming due to climate change, but not a novel one.

> Thank you very much for your sincere appreciation of the value of our manuscript. Concerning novelty, we did not want to claim that the used data or the used methods are by itself novel. Yet it is the first on that extends the data considered in the analysis and that combines this data with modelling efforts in an attempt ot explain the observed debris-flow propagation. Allow us to quote reviewer #2: *'By integrating contemporary methods, this study helps bridge knowledge gaps and contributes meaningfully to disaster science by reassessing historical events with improved precision and understanding'*. As there exist many modelling efforts for other debris-flow events in the past, we do no longer insinuate *novelty* but rather specify our contribtion more in detail.

> **ACTION:** We removed the notion of *novelty* from the manuscript. We attempt to better specify and emphasise the assets of this study.

Specific Comments:

1. There are grammatical errors and typos (such as line 2 (Abstract): instead of 'is' here will be 'are'). The authors should go through a thorough editing of the manuscript to correct grammatical errors and improve the readability of the paper, using a professional English editor as appropriate.

   > Thank you for pointing out this typo. We decided to follow the reviewer suggestion and pursued a thorough editing of the manuscript with focus on grammatical errrors and readability (also see answers to reviewers #1 and #2)

   > **ACTION:** Full manuscript editing as suggested.

1. The 30-m resolution Digital Elevation Models (DEM) generated from a 1:50,000 scale topographic map and aerial photographs have been used in this study. The volume of debris and water calculations of the 1987 avalanche using re-registering (resampling) of a 30-m resolution DEM is not accurate and precise. Therefore, the volume of debris and water is much higher than in the previous studies. Describe this limitation in the discussion section.

   Admittedly resolution is a limiting factor in the accuracy of the trigger volume estimate. We now expanded the discussion in this regard.

   **ACTION:** Added and rephrased discussion passage, particularly lines L390-L392 (new document).
   *'The absolute slab thickness, as inferred from DEM differencing, is certainly less reliable. The main reason is the 30-m resolution being rather coarse for the steep topography at the trigger site. Other parts of the uncertainty stem from difficulties in co-registration of the aerial photographs and from the fact that the reference DEM dates back to 1955.'*

1. The weather stations are sparsely distributed in the study areas. The weather parameters, such as temperature, precipitation taken from 40 to 50 km away from the areas where the volume of melt ice was calculated. The temperature within 40/50 km may vary several degrees. Therefore, this limitation should be explained in the discussion section.

   Thank you for raising this point. We admit that it is not ideal to only have station data from several tens of kilometre away. As our study region is located in mountainuous terrain, reliable extrapolation is far from evident. Yet, we do not attempt to accurately infer absolute values for certain atmospheric variables at a distant location. We rather use the remote stations (also reanalysis) to infer trends, distinct changes or outliers. Prominent examples are gradual and step increases in temperatures, extremes in seasonal snow-cover and warm-spell characteristics. We are confident that such relative comparisons hold over above distances.

   **ACTION:** We decided to start the discussion by pointing out that the presented atmospheric variables have to be interpreted with care (L352).
   *'We deliberately do not interpret absolute values of atmospheric variables neither in the climate reanalsyis nor in the distant station data. We limit ourselves to relative comparisons in terms of changes, trends and outliers.'*

1. This type of limitation is also observed in the hydrological data; the limitation of the hydrological data (discharge) should be explained in the discussion section.

The reviewer raises a fair point here. The only quantitative analysis of the discharge record is related to the estimationg of the total flood volume. Admittedly this value is prone to large uncertainties, certainly as we interprete measurements during extrems stream flow.

ACTION: A sentence was added in the discussion to discuss the limitation of this hydrological data.
*'[...] relies on streamflow observations that become delicate to interprete during extreme stream flow.'*

1. The caption of the figure should be numbered first, and then write the text of the caption, such as (a) Southern South America and (b) inset of the Metropolitan

    ACTION: Corrected as suggested - throughout the entire manuscript.

1. Technically, I think the method described in the manuscript is sound, but not innovative.

    We agree and refer the reviewer to our answer on his/her introductory comment (first comment).

Decision: Major revision

---

## Author Comment (AC3)

Point-by-Point Reply to Reviewer Reports on the manuscript entitled

**The Parraguirre ice-rock avalanche 1987, semi-arid Andes, Chile**

-

**A holistic revision**

by J.J. Fürst, D. Farías-Barahona et al.

**First of all, we want to thank all reviewers for the critical and useful comments on the manuscript. It was with great pleasure that we read their general appreciation of our work and the unanimous recommendation for publication. In the following, all review comments are repeated and addressed. Our responses are indented. Answers are given in blue and actions in orange. *Italic font* is enquoted and used to present newly added or modified text passages in direct answer to a specific comment.**

We are very delighted about the fact that all three reviews are positive and suppportive for publication in 'Natural Hazards & Earth System Sciences' (NHESS). Most comments have an editorial character. We tried to accomodate them to the best of our knowledge. If reviewer comments disagree, a compromise is suggested. Furthermore, reviewer #3 requested additional discussion of limitations of the employed methods (DEM differencing, climate data scarcety, ...). We accounted for these. In summary, we hope that the editor can continue to consider our manuscript for publication in NHESS.

**REFEREE #2**

Thank you very much for the opportunity to review the manuscript titled "The Parraguirre Ice-Rock Avalanche 1987, Semi-Arid Andes, Chile — A Holistic Revision."

This well-written manuscript offers a compelling and thorough reevaluation of a significant disaster event that occurred over three decades ago. The authors have effectively resurrected this event through the lens of modern analytical techniques and technologies, which were unavailable at the time of the original occurrence. By integrating contemporary methods, this study helps bridge knowledge gaps and contributes meaningfully to disaster science by reassessing historical events with improved precision and understanding.

> Thank you very much for these kind words and the high esteem for studies that reassess historical events aiming at improving our understanding of the flow progression. Your comment is deeply appreciated and encouraging us in our 'resurrection effort'.

> **ACTION:** No actions required.

I commend the authors for their comprehensive approach; however, I suggest a few minor revisions to enhance the manuscript's clarity and accessibility:

1. Abstract: Please briefly mention your methodological approach in the abstract to better inform readers about the basis of your analysis.

   > Thank you for this comment. You are right - we were rather vague on the specific methds and data that are now available to us.

   > **ACTION:** Rephrased respective sentence in the abstract as follows:
   > *'We therefore retrace the past event using data and techniques that are now at hand. These include historic topographic maps, aerial imagery, meteorological and hydrological records as well as multi-phase mass-flow modelling.'*

2. Manuscript Length and Precision: The manuscript is quite extensive, and at times, overly descriptive, which may hinder reader engagement.

   - The data section could be more concise while still retaining essential information.

     > Thank you for pointing us at this section. We were able to reduce the text passage from 45 to 30 lines without (hopefully) too much loss of information.

**ACTION:** Text passage was significantly reduced.

- The discussion, though critical, becomes lengthy and should be streamlined to maintain focus and readability.

  Again this appears to be a constructive comment. We could identify several passages in the discussion that showed redundancy with previous sections. Moreover and in accordance with a comment from Reviewer #1, a sub-subsection on water budgeting was separated to facilitate readability. With this, we truly hope that the text became more concise and can better maintain focus.

  **ACTION:** Complete overhaul of discussion section. Also see answer to Reviewer #1

- The conclusion can be more impactful if limited to two well-crafted paragraphs summarizing key findings and implications.

  Thank you for this comment. We succeeded to streamline this section and make the descriptions more concise. We could however not follow the suggestion to condense the conclusion in 2 impactful paragraphs. The reason is that reviewer #1 requested to add an extra outlook paragraph. We therefore kept the initial subdivisions.

  **ACTION:** A copy-edit of the conclusion section was employed that resulted in a text reduction of about 20%. Another paragraph was added on request of reviewer #1.

Overall, this is a valuable and timely contribution, and with these refinements, the manuscript will become even more effective and reader-friendly.